# Associations between Frailty and Ambient Temperature in Winter: Findings from a Population-Based Study

**DOI:** 10.3390/ijerph20010513

**Published:** 2022-12-28

**Authors:** Fenfen Zhou, Wensu Zhou, Wenjuan Wang, Chaonan Fan, Wen Chen, Li Ling

**Affiliations:** Department of Medical Statistic, School of Public Health, Sun Yat-sen University, Guangzhou 510275, China

**Keywords:** ambient temperature in winter, cold, frailty, cross-sectional analysis, older adults, China

## Abstract

Frailty is an accumulation of deficits characterized by reduced resistance to stressors and increased vulnerability to adverse outcomes. However, there is little known about the effect of ambient temperature in winter on frailty among older adults, a population segment with the highest frailty prevalence. Thus, the objective of this study is to investigate the associations between frailty and ambient temperature in winter among older adults. This study was based on the Chinese Longitudinal Healthy Longevity Survey (CLHLS) of older adults aged ≥65 years from the 2005, 2008, 2011, and 2014 waves. The 39-item accumulation of frailty index (FI) was used to assess the frailty status of the participants. The FI was categorized into three groups as follows: robust (FI ≤ 0.10), prefrail (FI > 0.10 to <0.25), and frail (FI ≥ 0.25). Generalized linear mixed models (GLMMs) were conducted to explore the associations between frailty and ambient temperature in winter. A generalized estimating equation (GEE) modification was applied in the sensitivity analysis. A total of 9421 participants were included with a mean age of 82.81 (SD: 11.32) years. Compared with respondents living in the highest quartile (≥7.5 °C) of average temperature in January, those in the lowest quartile (<−1.9 °C) had higher odds of prefrailty (OR = 1.35, 95% CI 1.17–1.57) and frailty (OR = 1.61, 95%CI 1.32–1.95). The associations were stronger among the low-education groups, agricultural workers before retirement, and non-current exercisers. Additionally, results from the GEE model reported consistent findings. Lower levels of ambient temperature in winter were associated with higher likelihoods of prefrailty and frailty. The findings on vulnerability characteristics could help improve public health practices to tailor cold temperature health education and warning information.

## 1. Introduction

Frailty is a consequence of accumulated physical, psychological, and social deficiencies with loss of reserves and reduced resistance to stressors during the aging process [1,2,3]. It has been shown that frailty is associated with increased risks of falls [4], disability [5], hospitalization [6], and death [7]. The frailty index (FI) is one of the most commonly used measures of frailty [8]. To capture an individual’s cumulative health deficits, most studies calculate the FI through standard comprehensive geriatric assessments [9]. Although studies using this approach usually do not include the same number or type of indicators to estimate frailty, it is shown that random selection of variables can produce comparable results [10]. According to a recent meta-analysis, approximately 43% and 10% of participants aged over 65 years are prefrail and frail, respectively, in China based on 14 studies with a sample size of 81,258 [11]. Worldwide, it is estimated that the prevalence is 41.6% for prefrailty and 10.7% for frailty among elderly adults [12].

Many studies have revealed that frailty is a complex and chronic process which could be triggered by multiple genetic and acquired factors [13], and the latter include aging [14], socioeconomic status [15,16], and lifestyle factors [17,18]. With an advancing understanding of the risk factors for frailty, environmental factors have attracted increasing attention in recent research, one of them being temperature [19]. Especially cold ambient temperature, as an important indicator in the field of environment and health, has a great impact on the health of the elderly [20]. A growing number of studies revealed that cold ambient temperature could increase the risk of specific diseases such as cardiovascular and respiratory diseases, and other studies have reported significant associations between cold ambient temperature and mortality in older adults [21,22]. Furthermore, several earlier studies have reported the adverse effects of cold ambient temperature on physical function [23]. For instance, based on a cross-sectional experimental study involving 88 older women (mean age 78 years), it was found that the physical performance of participants was worse in a moderately cold (15 °C) climate chamber compared with that in a warm/normal (25 °C) climate chamber [24]. One Japanese study enrolling 67 older people aged from 66 to 93 years indicates that cold temperature is an independent determinant of the change in physical performance measured by a self-designed scale [25]. Cold ambient temperature might be related to the deterioration of mental health, especially for individuals with poor mental status [26]. In recent work, cold ambient temperature has also been shown to be a potential cause of cognitive decline [27]. Indeed, some explanations could be used to support these findings. First, the tolerance of the elderly of hypothermia is more limited. Therefore, the elderly would feel uncomfortable under the influence of cold ambient temperature [28]. Biologically, cold ambient temperature could affect human body systems, such as the circulatory, respiratory, and digestive systems [29], and the duration of cold effects is longer than that of hot effects [30,31]. Although previous studies have suggested the significant linkages between population health and cold ambient temperature, these findings merely provided indirect evidence to support the associations between frailty and cold ambient temperature. This is because that frailty is viewed as a complex system combined with a comprehensive physical and psychological performance status rather than one aspect of health [32]. Hence, the impact of cold ambient temperature on the development of frailty still needs further study.

With the success of medicine and better nutritional support, population aging is dramatically accelerating worldwide [33]. Based on World Population Prospects 2022, the global population would reach eight billion, of which the proportion of older people aged 65 years and over was projected to rise from 10 percent in 2022 to 16 percent in 2050 [34]. However, the prevalence of frailty is substantially higher among the elderly population [35]. Moreover, aging is a multifactorial process characterized by disorder and loss of function at multiple levels and systems [36], which collectively results in vulnerability to environmental exposure [37]. The growing burden of frailty on the elderly could pose major economic challenges to the long-term care system in the coming decades [38]. Some studies found a positive correlation between indoor and outdoor temperature, but indoor environments are influenced by participant heating behaviors and building characteristics [39,40]. Given the scarcity of indoor temperature, using outdoor measurements has been common practice in the study of climate and health. These outdoor temperatures are often measured at regional weather stations in the open field to eliminate influences of coincidental surroundings.

Therefore, the present study aimed to investigate the associations between frailty and ambient temperature in winter among the elderly aged 65 years and over. Simultaneously, we aimed to further explore the modifying effects of ambient temperature in winter on frailty and identify the susceptible groups, which may help in designing and implementing more effective guidance for medical resource allocation and the care of older adults, especially in vulnerable populations.

## 2. Materials and Methods

### 2.1. Study Design and Population

Our study used data from the Chinese Longitudinal Healthy Longevity Survey (CLHLS), which is a national survey coordinated by the Center for Healthy Aging and Development at Peking University. CLHLS aimed to understand the factors affecting the health of the elderly (aged 65 or above), in particular the oldest-old (aged 80 or above) [41]. The project enrolled participants from 22 out of 31 provinces, municipalities, and autonomous regions of China, and these regions covered approximately 85% of the Chinese population. The first survey was started in 1998, and subsequent surveys were conducted in 2000, 2002, 2005, 2008, 2011, 2014, and 2018. In all waves of the CLHLS, about 113 thousand people have received face-to-face interviews, providing representative evidence to investigate determinants of healthy longevity in China. Key indicators collected include health status, disability, death and survival, demographic, family, socioeconomic, income level, and behavioral risk factors associated with mortality and healthy aging [42]. In the CLHLS, questionnaires were collected by trained staff with older adults themselves or their relatives or caregivers. The subsequent surveys were collected at 2–3-year intervals and were divided into two categories: one was for the survivors and the other was for the relatives of the deceased [43]. To reduce the attrition brought by death and loss, new participants were recruited for each survey based on similarities in sex, age, and general characteristics. The detailed response rate for each wave is not reported. According to the CLHLS, the overall interview rejection rate was 3.0% in 2005 and 4.5% in 2008 [44]. Therefore, the CLHLS has a high response rate among participants.

The current study collected information from 2005, 2008, 2011, and 2014 waves of the CLHLS because of the availability of residential address information (at the county or city level) and items used for creating the Frailty Index (FI). On this basis, we included participants aged ≥ 65 years who had completed surveys in two or more waves between 2005 and 2014 and without missing values on temperature information [45]. A total of 12,181 participants were excluded if they were lost (*n* = 4298) or had died (*n* = 7883) after the first survey, leaving a final analytic sample of 9421 participants.

### 2.2. Frailty Assessment

To measure frailty status, we calculated the FI by a standard procedure. Following the established research [46,47], FI comprised 39 indicators including self-reported health status, interviewer-rated health status, cognitive function, mental health, activities of daily living (ADL), instrumental activities of daily living (IADL), auditory and visual ability, heart rhythm, chronic diseases (e.g., hypertension, diabetes, heart diseases, stroke, etc.), and serious illness requiring hospitalization or being bedridden. Based on the questionnaire, cognitive function was measured by the Chinese version of the Mini-Mental State Examination (MMSE) with a total score of 30. Respondents with scores below 24 were denoted as having a cognitive impairment; the validity and reliability of the Chinese MMSE have been verified [48,49]. More details about the definition of FI using 39 indicators of various dimensions are listed in Appendix A. Each health indicator was dichotomous or ordinal, mapped to the interval of 0–1 to represent the severity of health deficits (e.g., for the self-reported health, “very good” was coded as 0, “good” as 0.25, “average” as 0.5, “bad” as 0.75, and “very bad” as 1). As the practices of the previous CLHLS study suggested, participants who two or more times suffered from a serious illness or were bedridden in the past 2 years were scored as 2 [50]. FI was calculated using unweighted counts of the actual number of deficits divided by the total possible number of deficits (see Formula (1)).
(1)$$FI=\frac{\sum_{i=1}^{n}k_{i}}{n}$$

In the equation, *k_i_* represents the value of the *FI* entry corresponding to the *i*_th_ index. *k_i_* = 0 means that the i_th_ index is completely healthy, *k_i_* = 1 indicates the health defects for the *i*_th_ index, and n represents the number of variables. For each participant, the *FI* scores were measured as the sum of deficit scores divided by the amount of deficit included, ranging from 0 to 1. It was a continuous variable, with a higher value indicating more severe frailty. Furthermore, the *FI* calculated by the proportion of the number of health deficits in this study was comparable to that in other studies. We categorized the continuous *FI* into three frailty statuses based on previous studies: robust (*FI* ≤ 0.10), prefrail (*FI* > 0.10 to <0.25), and frail (*FI* ≥ 0.25) [51].

### 2.3. Ambient Temperature in Winter Assessment

We collected information on the average temperature in January accurate to the residential units (i.e., at the county level) to reflect the levels of ambient temperature in winter, which was extracted from the community investigation datasets of the CLHLS. The CLHLS community datasets are auxiliary to the follow-up datasets of CLHLS, which were collected by the Center for Healthy Aging and Development Studies (CHADS) of National School of Development at Peking University from all kinds of publicly issued statistical yearbooks in China. The CLHLS community datasets contain information about the geographical environment, population, economic conditions, social welfare, and so on, of where the elderly respondents are living [19]. Two types of definitions for average temperature in January were used in our study, including absolute level (i.e., the average temperature in January) and relative level. Especially, the relative level of average temperature in January was considered as temperature change, that is, the difference between the average temperature in January of this year and last year, which was obtained from the National Earth System Science Data Center (http://www.geodata.cn (accessed on 20 October 2020)). It was pointed out that people have different degrees of adaptation to local climate conditions. People who live in warm areas are more susceptible to low temperatures [52,53]. Thus, we would like to further complement the link between temperature change and frailty in the elderly. Note that CLHLS did not provide the exact address of participants due to privacy protection, but it provided some residential community information, like gross domestic product (GDP), which could be used to identify the residential city in the dataset via cross-referencing the tables of the China City Statistical Yearbook [54]. The absolute and relative levels of average temperature in January were collected in each survey in accordance with multiple measurements of the *FI*.

### 2.4. Covariates

We controlled for the following previously proposed covariates that might be associated with frailty [48,55]. Demographic characteristics included age group (65–79 or ≥80 years, i.e., the oldest-old), sex (men or women), ethnicity (Han or Minority), current marital status (married or not married—widowed/divorced/single), and geographic regions (south China or north China). Socioeconomic status variables included residence (urban or rural), education (literate with at least 1 year of formal schooling or illiterate with no years of formal education), main occupation before the age of 60 (agriculture or non-agriculture), and family income last year (<10,000 or ≥10,000 RMB). Health behaviors consisted of smoking at present (yes or no), drinking at present (yes or no), exercising at present (yes or no), and social and leisure activities measured by the performance of seven items including gardening, reading newspapers/books, raising domestic animals, playing cards/mah-jong, watching TV/listening to radio, engaging in organized social activities, and any other personal outdoor activities. The social and leisure index was a continuous variable ranging from 0 to 7 with 1 point each if the participant answered yes to the questions. We further controlled for yearly rainfall (<800 or ≥800 mm) as a covariate at the city level [19]. In addition, data on sex, ethnicity, education, and main occupation before the age of 60 were collected from the first survey, other variables were collected in each survey.

### 2.5. Statistical Analysis

Descriptive analysis was conducted as mean with standard deviation (SD) or median with interquartile range (IQR) for continuous variables, and as numbers and percentages for categorical variables. With the amount of missing data being relatively high (24.5%) in the study, we performed multiple imputations of five to manage the missing values and calculated the *FI* after performing item-level imputations. Group differences were performed by using analysis of variance or the Kruskal–Wallis test for continuous variables and *χ*^2^ test for categorical variables.

To identify the potential linear or nonlinear relationship between average temperature in January both in forms of absolute and relative level and frailty, we employed generalized linear mixed models (GLMMs) with natural cubic spline items of different knots based on the minimum Akaike Information Criterion (AIC) value and visualized the exposure–response relationship between average temperature in January and frailty matched to each centigrade. Due to the two-level structure (i.e., repeated measurements nested within individuals) of the database, the random effect was included to address the within-subject correlation among the elderly. According to quartiles, average temperature in January was categorized into four groups (Q1: <P_25_, Q2: [P_25_~P_50_), Q3: [P_50_~P_75_), and Q4: ≥P_75_), and the highest quartile (Q4) was coded as a reference. We also classified average temperature in January in decrements of 1 °C as a continuous variable. The estimation of the odds of frailty was conducted using average temperature in January in absolute level and temperature change (i.e., the relative level), respectively. Two adjustment models were formulated: the age-adjusted model and the fully adjusted model. Compared with the former, the fully adjusted model further controlled for the covariates of demographic characteristics, socioeconomic status, and health behaviors. We calculated the average temperature in January in quartiles and estimated the odds ratios (ORs) with its 95% confidence intervals (CIs) to present the associations. Furthermore, we conducted a subgroup analysis to evaluate whether the effect of average temperature in January on frailty differed by age groups, sex, ethnicity, residence, current marital status, education, occupation, family income last year, smoking, drinking, exercising, and geographic regions after adjusting for related covariates. Referring to the previous study, a two-sample test assessing the statistically significant difference in estimated ORs within each subgroup was performed using the point estimate and standard error (SE) [56,57].

Sensitivity analyses were conducted to check the robustness of the results. First, we applied generalized estimating equation (GEE) models to repeat the above analyses. Second, all participants with or without subsequent surveys were added to a logistic regression model for the first survey analysis. Third, equivalent analyses were carried out on the original data with missing values. We also calculated the Variance Inflation Factor (VIF) among variables included in the present study, and all of them were less than 5, indicating a low chance of multicollinearity. All descriptive and inference statistical analyses were carried out in SAS version 9.4. A two-sided *p*-value of <0.05 was considered statistically significant.

## 3. Results

### 3.1. Descriptive Analysis

The general characteristics of the study participants are summarized in Table 1. The mean age of 9421 participants was 82.81 (SD: 11.32) years, 60.13% were 80 years and older, 55.34% were women, and 59.13% lived in rural areas. The average temperature in January was divided into four quartile groups (Q1: <−1.9 °C, Q2: [−1.9~4 °C), Q3: [4~7.5 °C), and Q4: ≥7.5 °C), and the highest quartile (Q4) was coded as reference. The prevalence of frailty in the first survey was 18.68%. In the last survey, the prevalence had increased to 28.37%, with changing in FI from 0.12 (IQR = 0.14) to 0.14 (IQR = 0.19). The elderly aged over 80 years, and women participants were more likely to be reported with worse frailty. In addition, participants without subsequent surveys were relatively older (91.51 vs. 82.81 years) and more frail (0.24 vs. 0.12) compared to those with subsequent surveys after the first survey (Appendix A).

### 3.2. Associations of Frailty and Average Temperature in January

Table 2 provides detailed results from the generalized linear mixed model for the associations between average temperature in January and frailty taking account of covariates, including age. Generally, participants living in the lowest quartile of average temperature in January compared to those in the highest quartile had higher odds of becoming prefrail (OR = 1.35, 95% CI 1.17–1.57) and frail (OR = 1.61, 95% CI 1.32–1.95) in the fully adjusted model. Furthermore, Figure 1 where the model found a statistically significant deviation from linearity (*p* < 0.05) displays consistent findings that higher levels of average temperature in January were protective, reducing the odds of developing frailty. In addition, Figure 2 depicts the subgroup analysis of each 1 °C decrease in average temperature in January and frailty. Although their associations are presented in Table 2, we focused here on the hierarchical level. It was conformable to the results obtained through the entire population, and the associations among individuals who had no formal education, mainly engaged in agriculture-related occupations before the age of 60, and had no exercising habit at present were significantly greater (*p*-value for modification effect <0.05).

The reference point for average temperature in January was set at −16 °C (5th percentiles). Data are shown as ORs (95%CI) of being frail in the fully adjusted model, which further controlled for age, sex, ethnicity, residence, current marital status, education, occupation, family income last year, smoking at the present, drinking at the present, exercising at the present, social and leisure activity index, yearly rainfall, and geographic regions.

Data are shown as ORs (95%CI) of being frail in the fully adjusted models.

### 3.3. Associations of Frailty and Temperature Change of Average Temperature in January

Table 3 reports the effects of temperature change between average temperature in January of this year and last year on frailty. The linear relationship between temperature change and frailty was observed in Figure 3. There were significantly negative associations between temperature change and frailty, even after adjusting for age. Each 1 °C decrease in temperature change was related to 2% higher odds of frailty (OR = 1.02, 95% CI 1.00–1.04) in the fully adjusted model. Analogous associations were also observed in the quartiles. The median of temperature change was −0.1 °C, P_5_ = −5.9 °C, and P_95_ = 4.1 °C. Temperature change was categorized into four quartile groups as follows: Q1: <−1.6 °C, Q2: [−1.6~−0.1 °C), Q3: [−0.1~1.6 °C), and Q4: ≥1.6 °C, and the last group (Q4) was coded as reference. In the fully adjusted model, compared with those in the highest quartile, those in the lowest quartile had higher odds of frailty (OR = 1.16, 95% CI 1.02–1.31). The associations were attenuated but remained significant after further adjustment for other covariates.

The reference point for temperature change was set at 0 °C, divided into two groups of temperature rise and temperature drop. Data are shown as ORs (95%CI) of being frail in the fully adjusted models, which further controlled for age, sex, ethnicity, residence, current marital status, education, occupation, family income last year, smoking at the present, drinking at the present, exercising at the present, social and leisure activity index, yearly rainfall, and geographic regions.

### 3.4. Sensitivity Analysis

The results of the sensitivity analysis were consistent with our main findings. First, we used GEE models to predict odds ratios for prefrailty and frailty (Appendix A). Results from the models were attenuated but still statistically significant for average temperature in January. Second, we conducted a logistic regression model including all participants with or without subsequent surveys for the first survey analysis (Appendix A). The results adjusting for sample attrition also reported that with the lower average temperature in January, the odds of prefrailty and frailty increased. Third, analyzing original data that has not been imputed, the observed statistically significant associations remained detectable and did not alter the results substantially (Appendix A).

## 4. Discussion

In the present study, we examined the associations between ambient temperature in winter and frailty among older adults. Our findings suggested that lower levels of average temperature in January as indicators both in absolute and relative forms were associated with increased odds of frailty, which persisted even after adjustment for demographic characteristics, socioeconomic status, and health behaviors. Additionally, we found some evidence that the relationships were greater among the individuals with certain characteristics based on subgroup analysis.

Recently, the health status related to environmental temperature has attracted increasing attention. Our results identified the adverse impact of cold ambient temperature on frailty, which was supported by previous evidence focused on cold ambient temperature and health. For instance, a cross-sectional survey conducted in Japan, completed with 342 participants aged ≥65 years at a rehabilitation facility or their home, reports that compared with that of warm groups, older adults feeling cold during winter peaks had an increased likelihood of physical function decline [58]. Similarly, a Chinese study involving 16 large cities reports that cold temperatures were responsible for stroke mortality, and most of them were caused by the number of days when the temperature is below the optimal temperature [59]. However, previous studies mainly focused on one aspect of health, providing only indirect evidence to support the associations between frailty and cold ambient temperature. As a comprehensive health indicator, frailty reflects the states of both physical and psychological performance. In particular, the current study showed that compared with the highest quartile, living in the lowest quartile of average temperature in January was not only associated with higher odds of frailty among the elderly, but also with higher odds of prefrailty. Some explanations of the findings were that due to the high level of skin heat conduction and the reduction of reflective vasoconstriction, the elderly exposed to a cold environment may have higher risks of being unable to maintain core body temperature [60,61]. With advancing age, the comfort zone of older people is higher than that of young because the adaptive mechanism of temperature regulation becomes less efficient [62]. Moreover, exposure to cold ambient temperature alters the concentration of central catecholamines (DA, epinephrine, and norepinephrine) [63]. Alterations in levels of central catecholamines may produce significant decrements in mental and cognitive performance as brain regions such as the prefrontal cortex are reliant on these neurotransmitters for normal function [64,65]. Additionally, cold ambient temperature could affect muscle activity performance by a decrease in all chemical reactions and contractility of muscle fibers and would not encourage physical activities due to the poor tolerability [66,67], along with the decline in physiological reverse across many organ systems, which may lead to frailty in the elderly. Currently, the evidence for frailty and ambient temperature in winter is limited. According to our findings, even with exposure to one unit decrease in average temperature in January, the risk of frailty should not be overlooked. From the perspective of public policy, developing more detailed cold warnings may be effective in protecting the elderly.

Furthermore, we observed that the temperature change in average temperature in January was also associated with frailty among the elderly, though with a small effect estimate. There is a similar pattern in a study on the relationship between the number of daily outpatient visits for respiratory disease and air temperature—with the temperature at 11.14 °C (annual average temperature), the number of daily outpatient visits for respiratory diseases increased by 9.12% (95% CI 5.93–12.42%) for every decrease of 1 °C [68]. Another study reports that the neurologic outcome became more favorable as the monthly ambient temperature increased by 1 °C (adjusted OR = 1.006, 95% CI 1.002–1.010) [69]. The underlying mechanism as to why a temperature decrease could affect frailty is not clear enough. Previous studies illustrate that a temperature decrease could affect individual susceptibility to stimulate the occurrence of infection through influencing viral activity and transmission, altering vectors and the host immune response, and changing allergen disposition [70]. In addition, decreased temperature would increase blood coagulation and plasma viscosity and change peripheral circulation resistance and blood pressure, increasing the vulnerability of the circulatory system [71,72]. Aging plays an important role in the adverse effects of decreased temperature on frailty because of the declined physical function. Furthermore, poor health conditions and aging weaken older adults’ ability to adapt to environmental risk factors [73]. Nevertheless, further exploration in future research focused on this complex area is still required.

Our study indicated that the associations between each 1 °C decrease in average temperature in January and frailty were modified by several characteristics of the elderly. We found that participants who had no formal education were more likely to be reported with frailty. One study concluded that the health impact of cold temperature on individuals among the less educated was slightly greater than that of others [74]. The use of education as an approximation of socioeconomic status and potential lifetime income has been well documented [75]. The education level in our study population was relatively low, so they were less likely to better protect themselves from the influence of the cold by heating or taking protective measures [76]. People with lower educational levels (either because of lower income or lower health literacy) could find it more difficult to maintain healthier lifestyles (e.g., better diets), being more vulnerable to the effects of cold ambient temperature. Furthermore, we found that participants who mainly engaged in agriculture-related occupations before the age of 60 tended to be more frail than those in non-agriculture. In our study, most of the participants were engaged in agricultural work, therefore, they were probably more likely to spend their time outdoors and more easily suffered from cold exposure. We also observed that participants who had no exercising habit at present had more frailty than exercisers did. As shown in another study, regular exercise could blunt the physiological effects of cold exposure, which reduced the risks of heart disease [77]. Regular and sustained exercise reflects an increased ability to develop coping mechanisms to protect individuals from cold. These findings suggest that frailty improvement efforts should focus more on vulnerable elderly in order to use better adaptation measures to cope with potential health risks of cold ambient temperature, such as promoting health education, providing shared thermostat appliances, and establishing early warning systems, would be helpful.

Previous research linked cold outdoor environments in winter with worse physical performance [78] and higher mortality rates [79]. Our results highlighted that cold ambient temperature was an underlying risk factor for frailty, which may have significant medical implications for the prevention and care of frailty. Namely, informal and formal health caregivers should check, wherever it is appropriate, that older people are in the right environmental temperature. Raising the ambient temperature and adding warm equipment properly, especially for the renovation of nursing homes, may reduce specific care needs and improve the effectiveness of related treatments, which requires further investigation.

Our study has several strengths. First, this study added evidence of ambient temperature in winter and frailty in the context of aging. In particular, the study provided results of quantification via analyses for ambient temperature in winter in forms of both absolute and relative levels. Second, our study relied on a large population sample, and the participants were randomly selected from about half of the counties and city districts in 22 of 31 provinces in Mainland China, accounting for 60.1% who were the elderly and aged over 80 years who need the most care. Finally, a large set of covariates were considered to minimize the impact of confounding bias, including demographic characteristics, socioeconomic status, and health behaviors.

Some limitations of this study should be acknowledged. First, this research is ecological. It is difficult to accurately measure the temperature exposure of individuals because of privacy protection, and the temperature used in the study was estimated only at the county or city level. Second, similar to a previous cross-sectional analysis, the causal relationship between frailty and ambient temperature in winter was not concluded. Third, FI is a continuous variable and it was categorized into three statuses (robust, prefrail, and frail), which made the FI less sensitive and informative to explain the trajectory of health. However, continuous FI is less likely to misclassify the phenotype [80], and the cutoff points used in our study have been shown to be reliable in a previous study [81,82]. The classification of FI in our study would not bias relevance. Fourth, as we did not track occupancy nor the activities of participants in each building, we were not able to determine behavioral influences, including thermal shock, moving from a warm indoor environment to a cold outdoor one. Last, we lacked more accurate and detailed information to consider the influence of sunshine duration, wind and wind chill, rain, snow, and air pollutants such as PM2.5. More data will need to be collected and analyzed in the future.

## 5. Conclusions

According to our study, lower levels of ambient temperature in winter were associated with higher odds of prefrailty (OR = 1.35, 95%CI 1.17–1.57) and frailty (OR = 1.61, 95%CI 1.32–1.95) among the elderly. Moreover, we found the relationships were greater in the elderly who had no formal education, mainly engaged in agriculture-related occupations before the age of 60, and had no exercise habits at present than those in their counterparts. Our findings reinforced the evidence of the potential hazards of cold ambient temperature, which had important implications for policies and programs to help prevent or improve the frailty of older adults in the process of healthy aging. However, it could not indicate the clothing situations of participants, whether perceived temperature or other scales would have stronger or weaker associations that require further study.

## Figures and Tables

**Figure 1 ijerph-20-00513-f001:**
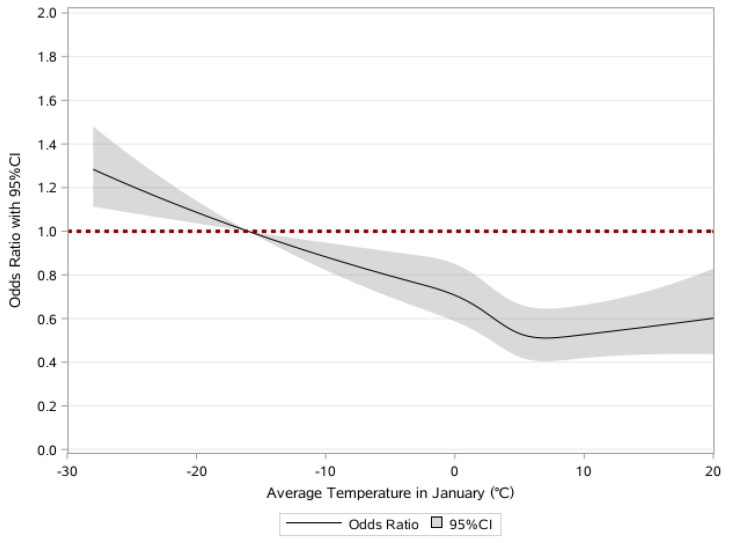
Restricted cubic splines for the associations between average temperature in January and frailty.

**Figure 2 ijerph-20-00513-f002:**
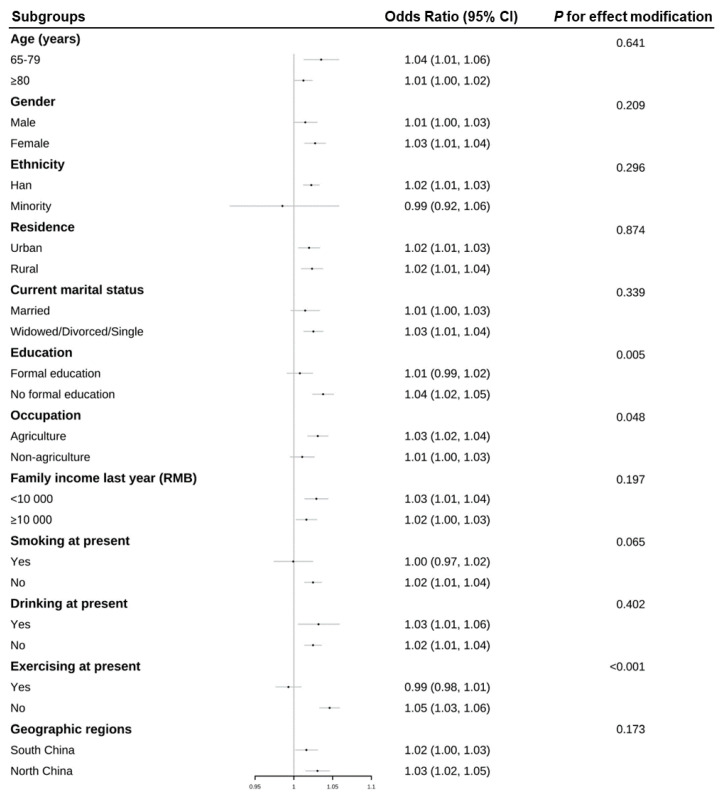
Subgroup analysis for the associations between each 1 °C decrease of average temperature in January and frailty.

**Figure 3 ijerph-20-00513-f003:**
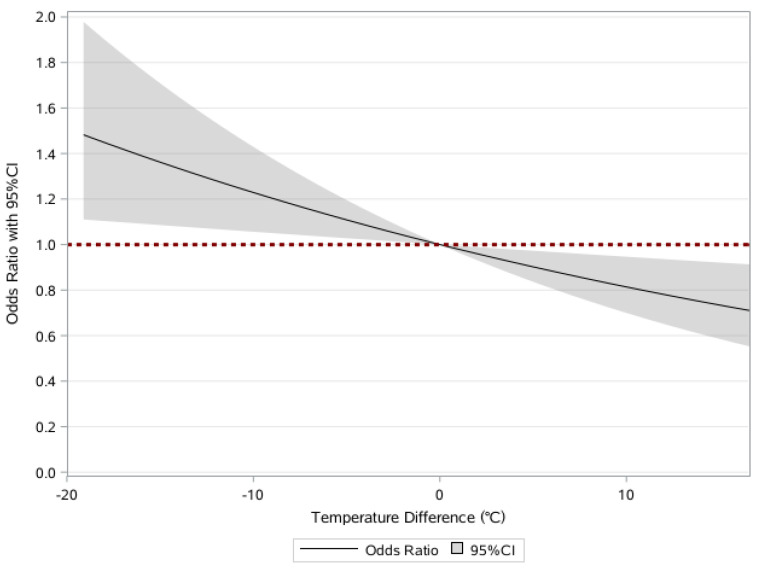
Restricted cubic splines for the associations between temperature change and frailty.

**Table 1 ijerph-20-00513-t001:** Characteristics of the study participants in the Chinese Longitudinal Healthy Longevity Survey (CLHLS).

Characteristics	Frailty Status				Total
	Robust	Prefrail	Frail	*p* Value	
*N* (%)	3843 (40.79)	3818 (40.53)	1760 (18.68)		9421
Age *, mean ± SD, years	77.29 ± 9.68	84.05 ± 10.42	92.17 ± 9.37	<0.001	82.81 ± 11.32
Age *, years, *N* (%)				<0.001	
65–79	2300 (59.85)	1263 (33.08)	193 (10.97)		3756 (39.87)
≥80	1543 (40.15)	2555 (66.92)	1567 (89.03)		5665 (60.13)
Sex *, *N* (%)				<0.001	
Man	2163 (56.28)	1575 (41.25)	469 (26.65)		4207 (44.66)
Women	1680 (43.72)	2243 (58.75)	1291 (73.35)		5214 (55.34)
Ethnicity, *N* (%)				0.878	
Han	3613 (94.02)	3612 (94.60)	1660 (94.32)		8885 (94.31)
Minority	230 (5.98)	206 (5.40)	100 (5.68)		536 (5.69)
Residence *, *N* (%)				0.015	
Urban	1542 (40.12)	1522 (39.86)	786 (44.66)		3850 (40.87)
Rural	2301 (59.88)	2296 (60.14)	974 (55.34)		5571 (59.13)
Current marital status *, *N* (%)				<0.001	
Married	2131 (55.45)	1403 (36.75)	339 (19.26)		3873 (41.11)
Widowed/Divorced/Single	1712 (44.55)	2415 (63.25)	1421 (80.74)		5548 (58.89)
Education *, *N* (%)				<0.001	
Formal education	2116 (55.06)	1461 (38.27)	431 (24.49)		4008 (42.54)
No formal education	1727 (44.94)	2357 (61.73)	1329 (75.51)		5413 (57.46)
Occupation, *N* (%)				0.650	
Agriculture	2449 (63.73)	2458 (64.38)	1093 (62.10)		6000 (63.69)
Non-agriculture	1394 (36.27)	1360 (35.62)	667 (37.90)		3421 (36.31)
Family income last year *, RMB, *N* (%)				0.002	
<10,000	2706 (70.41)	2730 (71.50)	1159 (65.85)		6595 (70.00)
≥10,000	1137 (29.59)	1088 (28.50)	601 (34.15)		2826 (30.00)
Smoking at present *, *N* (%)				<0.001	
Yes	1076 (28.00)	736 (19.28)	176 (10.00)		1988 (21.10)
No	2767 (72.00)	3082 (80.72)	1584 (90.00)		7433 (78.90)
Drinking at present *, *N* (%)				<0.001	
Yes	1039 (27.04)	744 (19.49)	209 (11.88)		1992 (21.14)
No	2804 (72.96)	3074 (80.51)	1551 (88.13)		7429 (78.86)
Exercising at present *, *N* (%)				<0.001	
Yes	1560 (40.59)	1380 (36.14)	311 (17.67)		3251 (34.51)
No	2283 (59.41)	2438 (63.86)	1449 (82.33)		6170 (65.49)
Social and leisure activity index *, median (IQR)	2.61 (2.06)	1.92 (1.88)	0.60 (1.59)	<0.001	2.00 (2.14)
Yearly rainfall *, mm, *N* (%)				<0.001	
<800	1289 (33.54)	1279 (33.50)	720 (40.91)		3288 (34.90)
≥800	2554 (66.46)	2539 (66.50)	1040 (59.09)		6133 (65.10)
Geographic regions *, *N* (%)				<0.001	
South China	2651 (68.98)	2621 (68.65)	1061 (60.28)		6333 (67.22)
North China	1192 (31.02)	1197 (31.35)	699 (39.72)		3088 (32.78)
Average temperature in January *, *N* (%)				<0.001	
Q1	851 (22.14)	863 (22.60)	485 (27.56)		2199 (23.34)
Q2	886 (23.05)	953 (24.96)	493 (28.01)		2332 (24.75)
Q3	1053 (27.40)	1072 (28.08)	449 (25.51)		2574 (27.32)
Q4	1053 (27.40)	930 (24.36)	333 (18.92)		2316 (24.58)

Abbreviations: *N* number; SD standard deviation; IQR interquartile range; Q1 1st quartile; Q2 2nd quartile; Q3 3rd quartile; Q4 4th quartile; * differences with statistical significance (*p* < 0.05) between the robust, prefrail, and frail groups.

**Table 2 ijerph-20-00513-t002:** GLMM analysis for the associations between average temperature in January and frailty.

Variables	Age Adjusted OR (95%CI)		Fully Adjusted OR (95%CI)	
	Prefrail	Frail	Prefrail	Frail
Quartiles of AT in January				
Q4	Reference	Reference	Reference	Reference
Q3	1.23 (1.12, 1.34) ***	1.29 (1.15, 1.45) ***	1.17 (1.07, 1.29) ***	1.14 (1.01, 1.30) *
Q2	1.27 (1.16, 1.40) ***	1.72 (1.53, 1.93) ***	1.22 (1.10, 1.36) ***	1.24 (1.07, 1.43) **
Q1	1.34 (1.22, 1.47) ***	2.26 (2.01, 2.54) ***	1.35 (1.17, 1.57) ***	1.61 (1.32, 1.95) ***
1-unit decrease of AT in January	1.01 (1.00, 1.01) ***	1.04 (1.04, 1.05) ***	1.01 (1.00, 1.02) *	1.02 (1.01, 1.03) ***

Abbreviations: AT in January, average temperature in January; Q1 1st quartile; Q2 2nd quartile; Q3 3rd quartile; Q4 4th quartile; 95%CI 95% confidence interval; OR odds ratio; note: In the fully adjusted model, ORs were adjusted for age, sex, ethnicity, residence, current marital status, education, occupation, family income last year, smoking at the present, drinking at the present, exercising at the present, social and leisure activity index, yearly rainfall, and geographic regions. Note that temperature change was not included in the fully adjusted model due to the high correlation with average temperature in January; *** *p* < 0.001, ** *p* < 0.01, * *p* < 0.05.

**Table 3 ijerph-20-00513-t003:** GLMM analysis for the associations between temperature change and frailty.

Variables	Age Adjusted OR (95%CI)		Fully Adjusted OR (95%CI)	
	Prefrail	Frail	Prefrail	Frail
Quartiles of AT in January				
Q4	Reference	Reference	Reference	Reference
Q3	1.05 (0.96, 1.14)	1.18 (1.06, 1.31) **	1.02 (0.93, 1.12)	1.08 (0.95, 1.22)
Q2	1.21 (1.11, 1.32) ***	1.38 (1.23, 1.54) ***	1.16 (1.06, 1.27) **	1.24 (1.09, 1.41) ***
Q1	1.20 (1.10, 1.31) ***	1.50 (1.34, 1.67) ***	1.08 (0.98, 1.19)	1.16 (1.02, 1.31) *
1-unit decrease of temperature change	1.02 (1.01, 1.03) **	1.05 (1.04, 1.07) ***	1.01 (0.99, 1.02)	1.02 (1.00, 1.04) *

Abbreviations: Q1 1st quartile; Q2 2nd quartile; Q3 3rd quartile; Q4 4th quartile; 95%CI 95% confidence interval; OR odds ratio; note: In the fully adjusted model, ORs were adjusted for age, sex, ethnicity, residence, current marital status, education, occupation, family income last year, smoking at the present, drinking at the present, exercising at the present, social and leisure activity index, yearly rainfall, and geographic regions. The temperature change was calculated by the average temperature in January of this year minus that of last year. Note that average temperature in January was not included in the fully adjusted model due to the high correlation with temperature change; *** *p* < 0.001, ** *p* < 0.01, * *p* < 0.05.

## Data Availability

The data used in this study were stored at Peking University (http://opendata.pku.edu.cn/ (accessed on 20 October 2022)). Researchers can obtain these data after submitting a data use agreement to the CLHLS team.

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
