# Peer review of "Associations between Frailty and Ambient Temperature in Winter: Findings from a Population-Based Study"

_ijerph, 2022, doi:10.3390/ijerph20010513_

Round 1

Reviewer 1 Report

There should be acknowledgement that there is a difference between indoor exposure to low temperatures and outdoor exposure. Those working outdoors (such as agricultural workers) and those intending to spend some time outdoors will/should wear appropriate protective clothing.  The indoor environment is affected by the thermal insulation and energy efficiency of the building.  There should also be acknowledgement that other outdoor (weather) factors could have a negative impact on frailty - wind and wind chill, exposed versus sheltered area, rain and/or snow. 

For non-outdoor workers, there is the possibility of thermal shock, moving from a warm indoor environment to a cold outdoor one.

These acknowledgements and recognitions of limitations should be made in the introduction, the discussion, and the conclusion.

Author Response

We would like to thank the reviewer for his time and valuable comments. These have considerably helped us improve our manuscript. We agree. Indoor temperature is a more exact predictor of exposure than outdoor temperature. Moderate correlations between indoor and outdoor temperatures suggest that indoor environments are influenced by participant heating behaviors [1]. Furthermore, outdoor effects on indoor climate are highly dependent on building characteristics such as building year, thermal insulation, dark walls and roofing, and living on higher floors of multi-story buildings [2]. Some studies found a positive correlation between indoor and outdoor temperature [3,4]. Given the scarcity of indoor temperature, using outdoor measurements has been common practice in the study of climate and health. These outdoor temperatures are often measured at regional weather stations in the open field to eliminate influences of coincidental surroundings. As we did not track occupancy nor the activities of participants in each building, we were not able to determine behavioral influences, including thermal shock. Estimating the proportion of time that participants spend indoors versus outdoors may reduce this error but be logistically infeasible for our large study population. Information on other outdoor weather factors was unavailable in the database, we were not able to control for confounders such as wind, rain, and snow exposure, which could affect the frailty status of the elderly. To better understand the relationships between frailty and ambient temperature in winter, it is necessary to acknowledge and recognize these limitations. We now have supplemented these aspects in the introduction, the discussion, and the conclusion sections.

  • [1] Sternbach T.J., Harper S., Li X., Zhang X., Carter E., Zhang Y., Shen G., Fan Z., Zhao L., Tao S., et al. Effects of indoor and outdoor temperatures on blood pressure and central hemodynamics in a wintertime longitudinal study of Chinese adults. J Hypertens. 2022;40:1950-1959. doi: 10.1097/hjh.0000000000003198.
  • [2] Basu R., Samet J.M. An exposure assessment study of ambient heat exposure in an elderly population in Baltimore, Maryland. Environ Health Perspect. 2002;110:1219-1224. doi: 10.1289/ehp.021101219.
  • [3] Nguyen J.L., Schwartz J., Dockery D.W. The relationship between indoor and outdoor temperature, apparent temperature, relative humidity, and absolute humidity. Indoor Air. 2014;24:103-112. doi: 10.1111/ina.12052.
  • [4] Tamerius J., Perzanowski M., Acosta L., Jacobson J., Goldstein I., Quinn J., Rundle A., Shaman J. Socioeconomic and Outdoor Meteorological Determinants of Indoor Temperature and Humidity in New York City Dwellings. Weather Clim Soc. 2013;5:168-179. doi: 10.1175/wcas-d-12-00030.1.

Reviewer 2 Report

Dear Dr. Jevtic,

Thank you for the opportunity to review this important manuscript titled, “Associations between frailty and ambient temperature in winter: findings from a population-based study”.

This manuscript reports findings of a study utilizing data from 9421 older adults who participated in the Chinese Longitudinal Longevity Survey (CLHLS), with the goal to investigate the link between frailty and ambient temperature in winter among this population. Frailty was assessed using the 39-item accumulation of frailty index (FI), and data was analyzed using generalized linear mixed models with appropriate adjustments. The authors found those living in the lowest quartile of average January temperature had higher odds of frailty (OR 1.61, 95% CI 1.32-1.95) and pre-frailty (OR 1.35, 95% CI 1.17-1.57). It was observed that associations were stronger among those in the low education groups, engaged in agricultural work prior to retirement, and non-current exercisers. The authors concluded that lower levels of ambient temperature in winter were associated with higher likelihoods of prefrailty and frailty.

The study design is well thought-out, methodology including statistical analyses rigorous and appropriate.  The paper itself is well-written, discussion tightly woven, incorporating relevant literature related to the topic. I would like to congratulate the authors on completion of this important study on a salient topic for older adults.  A minor comment: although causal relationship cannot be demonstrated by virtue of cross-sectional study design, the data from the relatively large sample size did present significant associational findings, which adds to the existing knowledge base and can form an important knowledge foundation from which further exploration can be galvanized in subsequent prospective cohort studies, as well as informing targeted public health interventions and anticipatory guidance/education campaigns.

Thank you again for the opportunity to review this excellent manuscript of this important study.

Author Response

We thank the reviewer for his time and comments, and his interest in our study.

Reviewer 3 Report

The topic of the paper is interesting, and it well Meet the publication scope of the journal. However, there are still some problems.

In Section 2.1, the Dataset is not cited.

In Section 2.1, please give the full spelling of FI.

In Section 2.1, more details on the survey should be added, such as the questionnaire and the statistics of the participants.

In Section 2.2, Table S1 cannot be viewed in the manuscript. The authors should give more detailed introduction on FI. And how to calculate the value of FI should be also added in the paper, and the specific calculation formula should be given.

In Section 2.3, why the authors select the average temperature in January to reflect the levels of ambient temperature in winter? Why not select the average temperature during December to Feburary?

In Section 2.3, why the authors use two types of definitions for average temperature in January? Why they use the difference between the average temperature in January of this year and last year? In Figure 3, the temperature difference ranges in -20~15℃, can the temperature difference in two years be so large?

In Section 2.3, please explain the reasonability of the source of the temperature data.

In Section 2.4, why the authors divide the age into two groups: 65-79 and 80+?

In Section 3.1, the Table S2 cannot be viewed in the manuscript.

The conclusion should be rewritten. Some quantitative results should be given.

Author Response

We thank the reviewer for his comments and good suggestions to improve this manuscript. We revised our manuscript and response to the suggested comments point-by-point as below.

Point 1: The topic of the paper is interesting, and it well Meet the publication scope of the journal. However, there are still some problems.

In Section 2.1, the Dataset is not cited.

Response 1: Thanks for your feedback and suggestion. Sorry for our negligence. Data were retrieved from the Chinese Longitudinal Healthy Longevity Survey (CLHLS) study. CLHLS was established in 1998 to explore the determinants of longevity in older adults. Samples were obtained from 22 of China’s 31 provinces, making this a nationally representative survey. The data quality of the CLHLS has been systematically evaluated [5,6]. We have referenced the dataset we used in the updated paper.

  • Gu D., Feng Q. Frailty still matters to health and survival in centenarians: the case of China. BMC Geriatr. 2015;15:159. doi: 10.1186/s12877-015-0159-0.
  • Zhu A., Yan L., Wu C., Ji J.S. Residential Greenness and Frailty Among Older Adults: A Longitudinal Cohort in China. J Am Med Dir Assoc. 2020;21:759-765. doi: 10.1016/j.jamda.2019.11.006.

Point 2: In Section 2.1, please give the full spelling of FI.

Response 2: Thanks for your feedback and suggestion. Sorry for our unclear description. We have now given the complete spelling of FI as “Frailty Index”.

Point 3: In Section 2.1, more details on the survey should be added, such as the questionnaire and the statistics of the participants.

Response 3: Thanks for your feedback and suggestion. Done as your comment. We have added more details to the survey. In the CLHLS, the questionnaires were collected by trained staff through face-to-face interviews with the older adults themselves or their relatives or caregivers. The follow-up questionnaires were divided into two categories: the questionnaire for interviews with the surviving participants and the questionnaire addressed to a close family member of the deceased interviewees [7]. Important measures included health status, disability, death and survival, demographic, family, socioeconomic, income level, and behavioral risk factors related to mortality and healthy aging [8]. CLHLS aimed to understand the factors affecting the health of the elderly (aged 65 or above), in particular the oldest-old (aged 80 or above). In the CLHLS, the detailed response rate of each wave is not reported. According to the illustration of the CLHLS, the overall interview refusal rate was 3.0% and 4.5% in the 2005 wave and 2008 wave, respectively [9]. Thus, the CLHLS has a high response rate among the participants.

  • Deng Y., Gao Q., Yang D., Hua H., Wang N., Ou F., Liu R., Wu B., Liu Y. Association between biomass fuel use and risk of hypertension among Chinese older people: A cohort study. Environ Int. 2020;138:105620. doi: 10.1016/j.envint.2020.105620.
  • Gu D. General Data Quality Assessment of the CLHLS. In Healthy Longevity in China: Demographic, Socioeconomic, and Psychological Dimensions, Zeng Yi, Dudley L. Poston, Denese Ashbaugh Vlosky, Danan Gu, Eds.; Springer Netherlands: Dordrecht, 2008; pp. 39-60.
  • Zeng Y. Towards Deeper Research and Better Policy for Healthy Aging --Using the Unique Data of Chinese Longitudinal Healthy Longevity Survey. China Economic J. 2012;5:131-149. doi: 10.1080/17538963.2013.764677.

Point 4: In Section 2.2, Table S1 cannot be viewed in the manuscript. The authors should give more detailed introduction on FI. And how to calculate the value of FI should be also added in the paper, and the specific calculation formula should be given.

Response 4: Thanks for your feedback and suggestion. Frailty is associated with an increased risk of death, falls, worsening disability, and hospitalization [10,11]. The frailty index (FI) is one of the most common instruments used to measure frailty [12]. To capture an individual's cumulative health deficits, most studies calculate the FI by a standard comprehensive geriatric assessment [13]. Our FI was the same as the previous CLHLS studies [14, 15]. FI included 39 self-reported items, including functional limitations, cognitive function, self-reported health status, interviewer-rated health status, mental health, auditory and visual ability, heart rhythm, and chronic diseases. Each item was dichotomous or ordinal, measured on a scale of 0 to 1 to represent the severity of health deficits. For example, self-rated health was scored as follows: very good = 0; good = 0.25; average = 0.5; bad = 0.75; or very bad = 1. We assigned a score of 2 if the respondent had had more than one serious illness in the past 2 years [16]. The FI was calculated using unweighted counts of the actual number of deficits divided by the total possible number of deficits. It was a continuous variable ranging from 0 to 1, with a higher value indicating severer frailty.

In the equation, ki represents the value of the FI entry corresponding to the ith index. ki =0 means that the ith index is completely healthy, ki =1 indicates the health defects for the ith index, and n represents the number of variables. For each participant, the FI scores were measured as the sum of deficit scores divided by the amount of deficit included, ranging from 0 to 1. All of the components were listed in Supplementary Table S1, which was uploaded in the supplemental materials. Sorry for our negligence, we have added an attachment at the end of the revised manuscript. We have now rewritten this aspect in the introduction and 2.2 section.

  • Clegg A., Young J., Iliffe S., Rikkert M.O., Rockwood K. Frailty in elderly people. The lancet. 2013;381:752-762. doi: 10.1016/S0140-6736(12)62167-9.
  • Dupre M.E., Gu D., Warner D.F., Yi Z. Frailty and type of death among older adults in China: prospective cohort study. BMJ. 2009;338:b1175. doi: 10.1136/bmj.b1175.
  • Rockwood K., Andrew M., Mitnitski A. A comparison of two approaches to measuring frailty in elderly people. J Gerontol A Biol Sci Med Sci. 2007;62:738-743. doi: 10.1093/gerona/62.7.738.
  • Searle S.D., Mitnitski A., Gahbauer E.A., Gill T.M., Rockwood K. A standard procedure for creating a frailty index. BMC Geriatr. 2008;8:24. doi: 10.1186/1471-2318-8-24.
  • Gu D., Feng Q. Frailty still matters to health and survival in centenarians: the case of China. BMC Geriatr. 2015;15:159. doi: 10.1186/s12877-015-0159-0.
  • Zhu A., Yan L., Wu C., Ji J.S. Residential Greenness and Frailty Among Older Adults: A Longitudinal Cohort in China. J Am Med Dir Assoc. 2020;21:759-765. doi: 10.1016/j.jamda.2019.11.006.
  • Goggins W.B., Woo J., Sham A., Ho S.C. Frailty index as a measure of biological age in a Chinese population. J Gerontol A Biol Sci Med Sci. 2005;60:1046-1051. doi: 10.1093/gerona/60.8.1046.

Point 5: In Section 2.3, why the authors select the average temperature in January to reflect the levels of ambient temperature in winter? Why not select the average temperature during December to Feburary?

Response 5: Thanks for your feedback and suggestion. Temperature information for December and February was unavailable in the database. However, January is the lowest mean temperature month in most areas of China, so the average temperature in January could reflect the ambient temperature in winter.

Point 6: In Section 2.3, why the authors use two types of definitions for average temperature in January? Why they use the difference between the average temperature in January of this year and last year? In Figure 3, the temperature difference ranges in -20~15℃, can the temperature difference in two years be so large?

Response 6: Thanks for your feedback and suggestion. In our paper, the average temperature in January was analyzed in the form of both absolute level and temperature change. Yes, without the latter, the results are clear and interesting. With it, those results are somewhat lost due to distraction by a complex side-issue. It was pointed out that people have different acclimatization to their local weather conditions. People residing in warmer areas were more susceptible to cold temperatures [17,18]. Thus, we would like to further complement the link between temperature change and frailty in the elderly. Temperature information from other years was not available in the database. Yes. The range of temperature change was from -20 to 15℃. In our study, the temperature change was categorized into four quartile groups as follows: Q1: <-1.6℃, Q2: [-1.6~-0.1℃), Q3: [-0.1~1.6℃), and Q4: ≥1.6℃. The median of temperature change was -0.1℃, P5=-5.9℃, and P95=4.1℃, with a small proportion of the samples having large temperature changes over two years. We have now supplemented the description of this aspect.

  • Group T.E. Cold exposure and winter mortality from ischaemic heart disease, cerebrovascular disease, respiratory disease, and all causes in warm and cold regions of Europe. The Eurowinter Group. Lancet. 1997;349:1341-1346.
  • Xiao J., Peng J., Zhang Y., Liu T., Rutherford S., Lin H., Qian Z., Huang C., Luo Y., Zeng W., et al. How much does latitude modify temperature-mortality relationship in 13 eastern US cities? Int J Biometeorol. 2015;59:365-372. doi: 10.1007/s00484-014-0848-y.

Point 7: In Section 2.3, please explain the reasonability of the source of the temperature data.

Response 7: Thanks for your feedback and suggestion. We obtained temperature data from the community questionnaire of the CLHLS in 2005, 2008, 2011, and 2014. The CLHLS community datasets are auxiliary to the follow-up datasets of CLHLS, which were collected by the Center for Healthy Aging and Development Studies (CHADS) of National School of Development at Peking University from all kinds of publicly issued statistical yearbooks in China. The CLHLS community datasets contain information about the geographical environment, population, economic conditions, social welfare, etc. of where the elderly respondents are living [19]. Furthermore, the community information provided in the CLHLS, like gross domestic product (GDP), could be used to identify the city of residence in the dataset via cross-referencing to tabulation in the statistical yearbook [20]. We have supplemented the description of this aspect in Section 2.3.

  • Zeng Y., Gu D., Purser J., Hoenig H., Christakis N. Associations of environmental factors with elderly health and mortality in China. Am J Public Health. 2010;100:298-305. doi: 10.2105/ajph.2008.154971.
  • Hu K., Keenan K., Hale J.M., Börger T. The association between city-level air pollution and frailty among the elderly population in China. Health Place. 2020;64:102362. doi: 10.1016/j.healthplace.2020.102362.

Point 8: In Section 2.4, why the authors divide the age into two groups: 65-79 and 80+?

Response 8: Thanks for your feedback and suggestion. Age as a risk factor for frailty in the Chinese elderly was reported in a previous study [21]. We would like to examine whether the association of ambient temperature in winter with frailty differed in the oldest-old (80 years and over). Therefore, following established research [22,23], the age in this study was divided into two groups: 65–79 and 80+.

  • Goggins W.B., Woo J., Sham A., Ho S.C. Frailty index as a measure of biological age in a Chinese population. J Gerontol A Biol Sci Med Sci. 2005;60:1046-1051. doi: 10.1093/gerona/60.8.1046.
  • Lv X., Li W., Ma Y., Chen H., Zeng Y., Yu X., Hofman A., Wang H. Cognitive decline and mortality among community-dwelling Chinese older people. BMC Med. 2019;17:63. doi: 10.1186/s12916-019-1295-8.
  • Zhang Y., Ge M., Zhao W., Liu Y., Xia X., Hou L., Dong B. Sensory Impairment and All-Cause Mortality Among the Oldest-Old: Findings from the Chinese Longitudinal Healthy Longevity Survey (CLHLS). J Nutr Health Aging. 2020;24:132-137. doi: 10.1007/s12603-020-1319-2.

Point 9: In Section 3.1, the Table S2 cannot be viewed in the manuscript.

Response 9: Thanks for your feedback and suggestion. Sorry for our negligence, Supplementary Table S2 was uploaded in the supplemental materials. We have added an attachment at the end of the revised manuscript.

Point 10: The conclusion should be rewritten. Some quantitative results should be given.

Response 10: Thanks for your feedback and suggestion. We agree. We have now rewritten the conclusion section with some quantitative results. This sentence has been revised as “lower levels of ambient temperature in winter were associated with higher odds of prefrailty (OR = 1.35, 95%CI 1.17-1.57) and frailty (OR = 1.61, 95%CI 1.32-1.95) among the elderly.” in the updated paper.